# Update on Management of Cardiovascular Diseases in Women

**DOI:** 10.3390/jcm11051176

**Published:** 2022-02-22

**Authors:** Fabiana Lucà, Maurizio Giuseppe Abrignani, Iris Parrini, Stefania Angela Di Fusco, Simona Giubilato, Carmelo Massimiliano Rao, Laura Piccioni, Laura Cipolletta, Bruno Passaretti, Francesco Giallauria, Angelo Leone, Giuseppina Maura Francese, Carmine Riccio, Sandro Gelsomino, Furio Colivicchi, Michele Massimo Gulizia

**Affiliations:** 1Cardiology Department, Big Metropolitan Hospital, 89129 Reggio Calabria, Italy; massimo.rao@libero.it; 2Cardiology Department, S. Antonio Abate Hospital, 54027 Trapani, Italy; maur.abri60@gmail.com; 3Cardiology Department, Ospedale Mauriziano Umberto I Hospital, 10128 Turin, Italy; irisparrini@libero.it; 4Clinical and Rehabilitation Cardiology Department, San Filippo Neri Hospital, ASL Roma 1, 00100 Roma, Italy; doctstefania@hotmail.com (S.A.D.F.); furio.colivicchi@gmail.com (F.C.); 5Division of Cardiology, Cannizzaro Hospital, 95121 Catania, Italy; simogiub@hotmail.com; 6Italy Cardiology Department, “G. Mazzini” Hospital, 64100 Teramo, Italy; laura.piccioni@aslteramo.it; 7Division of Cardiology, Department of Cardiovascular Sciences, University of Ancona, 60126 Ancona, Italy; cipollettalaura@gmail.com; 8Rehabilitation Cardiology Department, Humanitas Gavazzeni, 24125 Bergamo, Italy; bruno@passaretti.org; 9Department of Translational Medical Sciences, Federico II University of Naples, 80138 Naples, Italy; francesco.giallauria@unina.it; 10Cardiology Division, Annunziata Hospital Cosenza, 87100 Cosenza, Italy; angeloleone.al@gmail.com; 11Cardiology Complex Unit, “Garibaldi Nesima” Hospital, 95122 Catania, Italy; maurafrancese@virgilio.it; 12Division of Clinical Cardiology, ‘Sant’Anna e San Sebastiano’ Hospital, 81100 Caserta, Italy; carmine.riccio8@icloud.com; 13Cardio Thoracic Department, Maastricht University, 6202 AZ Maastricht, The Netherlands; sandro.gelsomino@maastrichtuniversity.nl; 14Heart Care Foundation, 50121 Florence, Italy; michele.gulizia60@gmail.com

**Keywords:** cardiovascular disease, women, gender, cardiovascular risk factors

## Abstract

Cardiovascular diseases (CVD) have a lower prevalence in women than men; although, a higher mortality rate and a poorer prognosis are more common in women. However, there is a misperception of CVD female risk since women have commonly been considered more protected so that the real threat is vastly underestimated. Consequently, female patients are more likely to be treated less aggressively, and a lower rate of diagnostic and interventional procedures is performed in women than in men. In addition, there are substantial sex differences in CVD, so different strategies are needed. This review aims to evaluate the main gender-specific approaches in CVD.

## 1. Introduction

Despite a lower prevalence of cardiovascular diseases (CVD) in women than men, the mortality rate and prognosis are poorer in females [1]. Women have been conventionally considered more protected, and, therefore, their real CVD risk has been largely underestimated [2,3,4,5]. As a result, less aggressive strategies are more likely to be used in women than men [2,4,5,6], as demonstrated by the lower rate of diagnostic and interventional procedures performed in females [2,4,6]. In addition, women are generally under-represented in most clinical trials [4,7]. Gender-related disparities in heart physiology have been widely demonstrated, leading to sex differences in CVD, which significantly influence different treatment strategies [4,8,9].

Therefore, CVD management should have a gender-specific approach that remains poorly applied in clinical practice. This study aimed to review main cardiovascular (CV) risk factors in women related to CVD and to discuss sex-specific treatment aiming at helping clinicians in adopting a more gender-specific clinical approach.

## 2. Cardiovascular Risk Factors in Women

The classic risk factors for CVD are comparable in women and men, but gender differences in the prevalence of each risk factor and unique factors exist for women (Figure 1). Indeed, smoking and dyslipidemia are more prevalent among men, whereas metabolic syndrome, sedentary, concomitant autoimmune, and chronic kidney diseases (CKD) are more frequent in women [5].

In Europe, data from the EUROASPIRE IV, a multi-centric study involving 7998 patients (24.4% females) referred to 78 centers in 24 countries for coronary heart disease (CAD), evidenced a poor risk factor management in coronary heart disease (CAD) in women than men [10,11].

According to these findings, the latest EUROASPIRE V survey (undertaken on 8261 CAD patients, 25.8% females) [12] showed a worse control of CV risk factors in women. On the contrary, a little gender gap in CV drugs intake has been evidenced [13].

Nevertheless, data analysis focused on gender differences in the patients’ awareness, showed a lower awareness about weight but a greater awareness about blood pressure (BP) and cholesterol target achievement in females [14].

### 2.1. Hypertension

Hypertension is the most usual modifiable risk factor for CVD, and lowering BP prevents morbidity and mortality in both sexes [15]. Premenopausal women usually have lower BP values than men [16]. However, after menopause, a steeper rise in hypertension rates is seen in women and about 80% of women aged ≥75 years have hypertension [17].

Females develop more often isolated systolic hypertension (ISH), reflecting aortic stiffness (AoS), and have a higher prevalence of strokes and heart failure (HF) with preserved ejection fraction (HFpEF) [18].

Hypertension is more frequently uncontrolled in women. Such types of hypertension are exclusive of women, such as hypertension related to oral contraceptive (COCs) use or hypertensive disorders during pregnancy (HPD) [19,20].

Several specific sex/gender factors could explain women’s unique arterial hypertension pathophysiology. Estrogens deficiency in post-menopause plays a crucial role in hypertension development due to adaptations of the sympathetic nervous system (SNS), renin-angiotensin-aldosterone system (RAAS), body mass (BM), endothelial function, oxidative stress, and salt sensitivity [20].

However, recent studies have also shown that differences in SNS, RAAS activation, sex chromosomes, and immune system, independently by the gonadal hormone status, contribute to the sexual dimorphism in BP control [21].

Nonetheless, there is currently no substantial evidence showing different efficacy of antihypertensive therapy based on gender.

In a large meta-analysis including 87,349 women, Turnbull et al. evaluated different BP-lowering regimens using similar cut-offs for men and women, showing equal protection against severe vascular complications in both sexes [22]. In this study, calcium channel blockers (CCBs) reduced the risk of stroke more than beta-blockers (BBs) or ACE inhibitors (ACEI) only in women, but not in men. However, CCBs did not differ from BBs, ACEI, or diuretics in protecting CAD, cardiac death, or death from any other cause in both genders.

Therefore, guidelines for managing arterial hypertension recommend no different BP targets or particular drug classes, based on the patient’s gender [23].

It’s well known that current guidelines suggest a more intensive treatment of hypertension to a goal systolic BP ≤ 130 mmHg, based on large trials such as SPRINT (Systolic Blood Pressure Intervention Trial). However, a prespecified subgroup analysis of this study failed to show a statistically significant benefit from the intensive treatment versus the standard therapy in women [24].

Further studies, including larger women population with hypertension, are needed to test the hypothesis for implementing more gender-specific treatment indications.

Therefore, to date, gender should not influence selecting antihypertensive therapies, apart from evaluating gender-specific side effects or contraindications in pregnancy [20].

Common side effects of antihypertensive therapy occur more frequently among women than men. ACEI-induced cough is twice as common in women than in men, and women are more likely to complain of CCBs related peripheral edema and to develop diuretic-induced hyponatremia and hypokalemia [20,25]. On the other hand, diuretics might positively affect the prevention of osteoporosis in postmenopausal women through reduced urinary calcium excretion [24,25].

#### Gender Differences in Hypertension-Related Target Organ Damage

It is well known that hypertension-mediated organ damage (HMOD) in vessels or organs (heart, brain, eyes, and kidney) is a marker of pre-clinical CVD associated with increased CV morbidity and mortality [26]. Therefore, knowledge of the presence of HMOD is of significant importance to better stratify CV risk and for the optimal management of hypertensive patients [26]. Several gender differences in HMOD have been described in the last years, and estrogens play a crucial role in HMOD pathogenesis [20].

Postmenopausal hypertensive women have more often ISH, reflecting an increase in AoS [27], more concentric left ventricular (LV) remodeling and less LV in response to arterial hypertension, resulting in a higher LV mass index and greater prevalence of HFpEF [18]. Moreover, in women, it has been demonstrated that the regression of hypertensive left ventricular hypertrophy (LVH) is more difficult to be obtained than in men, and residual hypertrophy is more common despite effective antihypertensive strategies and adequate BP control [28].

It is important to note that LVH has a well-demonstrated association with CV morbidity and mortality, and some studies have demonstrated that a higher LV mass index have a more significant impact on worse clinical outcome in women than in men [28].

Obesity, more prevalent in women than in men, may also potentiate the effect of hypertension on LVH in women, and the presence of increased body mass index (BMI) may be responsible, at least in part, for the lack of LVH regression as observed in the Strong Heart Study population [28,29].

Significant differences among male and female individuals were observed on vasculature damage, including arterial stiffness and intimate-medium thickness (IMT), carotid plaque size and compositions, and small arteries. Recently, extensive prospective studies showed that men have higher carotid IMT and carotid plaques than women at any decade of age. In contrast, women have less plaque burden, more stenosis, and a more positive remodeling of internal carotid arteries [30]. Intraplaque hemorrhage, a marker of plaque instability, is more frequent in men than women. Still, with increasing age, the probabilities of intraplaque carotid bleeding in women become closer to that of men [31]. It has been well assessed that arterial stiffness increases more significantly in women with aging, related to two-fold higher mortality than men [27].

Coronary microvascular dysfunction (CMD) leads to a significant increase in endothelial shear stress which negatively influences coronary anatomy and function and is strongly correlated with adverse CV events [32,33]. Moreover, the smaller coronary arteries size associated with a higher blood flow has been reported as a causal factor of a greater prevalence of CMD in women. In addition, the direct effect exerted by hypertension on microcirculation causes intramural arterioles’ remodeling and interstitial fibrosis. The reduction in microvascular density has also been involved in the development of CMD [32]. Nevertheless, estrogens have a protective role in premenopausal women [34]. The mechanism would seem due to early estrogen loss resulting in chronic activation of the RAAS [20]. Therefore the incidence of CMD significantly rises in postmenopausal women [32].

A more significant and earlier hypertension-related microvascular dysfunction in the female sex has been recently supported by the findings that the media/lumen (ML) ratio was higher in women than in men after correction for classical CV risk factors and age [35].

In contrast, microvascular obstruction areas (also knowns as “no-reflow”) following myocardial infarction (MI) remodeling are smaller in women and more presumably linked to distal atherothrombotic embolization, microvascular impairment, and reperfusion insult [36].

Microalbuminuria is a marker of CV and renal diseases, and it is a sign of HMOD in essential hypertension [37]. Irrespective of BP levels, microalbuminuria, urinary creatinine, and albumin excretion is lower in women [38]. Postmenopausal women have a more rapid deterioration of renal function, while BP control results in higher proteinuria lowering men than in women.

Experimental animal studies suggest a role for T regulatory cells and RAS system in sex differences in hypertensive kidney injury [39].

Hypertensive retinopathy (HR), which refers to retinal microvascular signs which develop in response to elevated BP, predicts stroke, congestive heart failure (CHF), and CV mortality, independently of traditional risk factors [40]. Hypertensive retinal vascular signs can be classified into arteriolar changes (narrowing of the retinal arteriolar vessels due to vasospasm and increased vascular tone, arterio-venous crossing or nicking, and arteriolar wall opacification), and more advanced retinal lesions (microaneurysms, retinal hemorrhages, cotton-wool spots, hard exudates, optic nerve ischemia, and optic disk swelling) [41].

HR is more prevalent in males than in women. This difference may be explained by differential distribution in risk factors [42]. It has been shown that antihypertensive therapy results in regression of HR and that this effect is mainly due to BP reduction and rather than antihypertensive drugs [43].

### 2.2. Diabetes Mellitus

Diabetes mellitus (DM) is estimated to affect over 13 million women in the United States, with 90–95% having type 2 diabetes (T2DM) [43].

It has been observed that people with T2DM have a 2–3 times higher CV risk than people without diabetes [44]. Therefore preventing microvascular complications could reduce major adverse CV events, as far as T2DM, is involved in CAD development, in plaque burden, in the lesion extent and vascular remodeling, hesitating in a severe and diffuse coronary artery narrowing [45,46].

Regarding females, it has been assessed that diabetes significantly attenuates premenopausal cardioprotection [47]. Women with T2DM rise a greater CV risk compared with non-diabetic women and diabetic men both [47]. A greater risk of CVD mortality in diabetic women compared with men has also been reported [47]. A more pronounced hypercoagulability, endothelium dysfunction, and metabolic and cellular alterations resulting in functional and structural abnormalities are involved in the mechanism of myocardial dysfunction with a poor impact in women [43]. In addition, an enhanced risk of HF as well as HF mortality has been long been recognized in women with T2DM compared with men [47] (Figure 2).

The T2DM pathogenesis is strictly linked to obesity. Thus, it is known that the BMI, adipose tissue dysfunction and the expression of adipokines secreted by the adipose tissue play an essential role in the T2DM etiopathogenesis [29,48]. Since all these features significantly differ between men and women, sex differences are particularly relevant in T2DM [48,49]. To be more specific, obesity is more prevalent in women [50], whereas in men there is a higher risk of developing T2DM [51]. However, accumulating visceral fat is linked to the development of T2DM [52].

Moreover, insulin sensitivity is more frequently observed in women [53]. However, the progressive loss of estrogen production during aging slowly results in significant changes in body shape, increasing abdominal fat storing, and shifting from the gynoid to the android shape [54]. Finally, CV relative risk seems to be more strongly correlated to T2DM in women [55].

After these considerations, we may state that an aggressive approach for diabetic patients is required in both sexes. A lighter treatment based on their hypothetic, more favorable hormones profile is no longer acceptable [55].

### 2.3. Cholesterol

A higher prevalence of elevated total cholesterol (TC) levels and lower high-density lipoprotein cholesterol (HDL-C) values have been shown in women than in men [56]. In Italy, in 2008–2012, the levels of total TC and low-density lipoprotein cholesterol (LDL-C) were lower in men, with a prevalence of 65% of TC > 200 mg\dL, compared to the 69% of women [57].

The 2018 ACC/AHA cholesterol guidelines identified during early menopause a rise in LDL and total cholesterol with increased CV risk [19].

Although a strong correlation between menopause and changes in cholesterol levels has been previously described, a more precise assessment of the significant increase in TC, LDL-C, and Apolipoprotein B (ApoB) levels occurring in the final menstrual period (FMP) has been well recognized in SWAN study(Study of Women’s Health Across the Nation) [58,59]. Additionally, relevant variations in carotid plaque burden have also been observed in the follow-up [59,60].

In the Tromso Study, an association between carotid atherosclerosis and earlier menopausal age was reported [61]. Finally, premenopausal values of LDL-C, HDL-C, and triglycerides have been well identified as strong predictors of carotid IMT in the postmenopausal phase in the Pittsburgh Healthy Women Study [62].

Therefore, the phase between one year before and one year after FMP should be considered as the critical time for lipid profile changes. As a consequence, a more careful lipid monitoring approach in premenopausal and perimenopausal women should be performed [47].

The INTERHEART TC/HDL-C. The INTERHEART study has investigated the Apolipoprotein B (ApoB)/ Apolipoprotein A1 (ApoA1) and TC/HDL-C ratios finding an association with acute myocardial infarction (AMI) more frequently in women than in men [63].

Statins have an indication in secondary prevention without difference of gender for major CV events [64]. In recent years, the aggressive reduction of LDL cholesterol with the Proprotein Convertase Subtilisin/Kexin type 9 (PCSK9 inhibitors) contributed to the significant reduction in ischemic events without apparent gender differences [65].

### 2.4. Smoking

A recent World Health Organization (WHO) report on the global tobacco epidemic showed that, in 2013, 19% of women and 38% of men aged 15 years old and above smoked tobacco in the WHO European Region. This average among European women is sensibly higher than those observed in the WHO African, South-East Asia, Eastern Mediterranean and Western Pacific Regions (2–3%) [65]. The INTERHEART study reported that smoking had a similar risk of AMI in both genders [66]. An increase in tobacco or e-cigarette smoking has been documented in the last years, contributing to a 25% increase in CV risk [67].

Consequently, the prevalence of women smokers is becoming higher than men, impacting morbidity and mortality connected to smoking-related diseases [68].

### 2.5. Obesity

Obesity rates are rising worldwide, involving about a third of the world’s population [69]. 

In the WHO European Region, the age-standardized prevalence of obesity in 2016 was 21.85% for men and 24.46% for females, with an increasing parallel trend in two genders [68].

A higher prevalence of obesity in women (18%) compared to men (10%) occurs [70]. Moreover, obesity in pregnancy may contribute to the development of hypertension and GD [71].

BMI is commonly used to define overweight or obese patients, although either important is the fat localization [72]. According to data from the Framingham Heart Study, the excess risk of cardiovascular disease CVD attributed to obesity after adjustment for waist circumference was 64% in women versus 46% in men [73]. In a study by Chen et al. including 2863 postmenopausal women, trunk fat was strongly associated with CV risk despite a normal BMI [74]. Central obesity is more common in women than men contributing to metabolic syndrome (MS), especially in postmenopausal women [75] (Figure 3). According to data from the Framingham Heart Study, the excess risk of CVD attributed to obesity after adjustment for waist circumference was 64% in women versus 46% in men [73].

### 2.6. Physical Activity

By the pooled data from 358 surveys on physical activity across 168 countries, including 1.9 million population between 2001 and 2016, authors reported a higher rate of physical inactivity in women than men (31.7% versus 23.4% in 2016) [76].Young women are less engaged in physical activity (PA) than men, with a continuing decrease over the years leading to increased risk of CV disease [43,77]. Extending the duration of physical activity beyond 10 minutes in older people is essential for staying healthy [78].

### 2.7. Chronic Kidney Disease

In addition to the conventional CV risk factors, CKD is strongly associated with CV events [79]. An enhanced prevalence of CKD in women, including primary injuries and secondary involvements in systemic diseases, has already been described [80,81] Women who have a longer life expectancy because of their age, have a greater reduction in the glomerular filtrate rate (GRF), which could be a potential cause of the more CKD_s_ prevalence in females. Additionally, it has been hypothesized that a significant role of sex hormones is involved in gender disparities [82].

Furthermore, it has been shown that women are more likely to be affected by autoimmune diseases like Systemic Lupus Erythematosus (SLE) occurring in their childhood [83,84]. Therefore, SLE-related nephritis (Lupus Nephritis, LN), has been reported in more than 75% of SLE patients [85]. LN, resulting from autoimmune mechanisms, may lead to kidney failure [84]. Furthermore, Systemic Sclerosis (SS), prevalently affecting women, could determine a kidney impairment in 5% of patients [86]. In addition, pyelonephritis, more common in women for anatomic features, can lead to CKD over time [87].

Moreover, a significant relationship has been found between pregnancy and CKD. It seems to be related to complex anatomic-functional modifications of kidneys that characterize maternal physiopathology. Both Acute Kidney Injury (AKI) and preeclampsia (PE) can be pregnancy-related complications leading to the development of CKD [88]. AKI represents a preeminent problem significantly increasing maternal and fetal morbidity and mortality [88]. Moreover, AKI often results in CKD and end-stage kidney disease (ESKD). Consequently, prompt recognition and suitable treatment for AKI are mandatory during pregnancy [88].

Preeclampsia occurs in 5–8% of pregnancies causing 15–20% of pregnancy-related AKI representing a potential cause of intrauterine and perinatal mortality, preterm delivery, and intrauterine growth restriction (IUGR) [88]. 

Conversely, a higher risk for AKI and CKD progression has been reported in men while a lower incidence of renal replacement therapy (RRT) has been reported in women [89,90]. Finally, males are also more likely to receive kidney transplants than females [90].

### 2.8. Anemia

Anemia is largely diffused among the general population. It is significantly influenced by economic status and consequently by nutritional deficiencies representing a worldwide health problem. Despite its gender-balanced spread, anemia is particularly common in women [91], with a prevalence of 38% during pregnancy and 29% in non-pregnant women [92].

Iron deficiency (ID) is the preeminent cause of anemia in females with an incidence ranging between 15 to 18% [91]. The etiology of ID in women is multifactorial [93]. Slow bleeding from uterine fibroids, heavy menstruation [94], intrauterine devices (IUDs), and other gynecological conditions have been considered as causative factors [92,95]. Moreover, hemoglobinopathies, gastrointestinal (GI) bleedings, autoimmune diseases, kidney failure, parasitosis, other nutritional deficiencies (such as vitamin B12, folate), acute and chronic diseases, and malabsorption are the other most common general causes of anemia [92,93,96].

Adverse maternal and fetal events have been reported as a consequence of anemia [91,93].

The WHO established that a reduction of 50% of anemia in fertile females is part of the six global nutritional goals to be achieved by 2025 [97], setting the cut-off hemoglobin (Hb) concentration at <110 gl and <120 gl for pregnant women and non-pregnant women, respectively [98]. Vegetarian or vegan diets [99], younger and older maternal age [100], multiple pregnancies [101], and previous anemias are predisposing risk factors of developing ID in pregnancy.

If anemia has been confirmed and other causes of bleeding have been excluded, gastroscopy and colonoscopy should be recommended [102]. Finally, a higher need for transfusion after surgery has been reported in women undergoing surgery [91]. Oral or intravenous iron supplements are the recommended treatment after the exclusion of removable causes [91].

## 3. Cardiovascular Gender-Specific Risk Factors

### 3.1. Hypertensive Disorders during Pregnancy (HPD)

HPD including chronic hypertension, gestational hypertension (GH), and preeclampsia (PE) complicate 5% to 10% of pregnancies. They are associated with increased maternal and perinatal morbidity and mortality and a higher risk of developing post-partum hypertension and long-term CVD [103,104]. Accordingly, the latest European Society of Cardiology (ESC) Guidelines for managing arterial hypertension recommend a careful follow-up to assess BP and metabolic disorders in women with a history of HPD [23]. All antihypertensive medications cross the placenta, and no large-scale study in pregnant women has compared the use of one antihypertensive drug class to another [105]. Methyldopa, labetalol, and CCBs are the drugs of choice to treat hypertension in pregnancy. Medications to avoid (Class III) during pregnancy are ACEI, angiotensin receptor blockers (ARBs), and nitroprusside because of the risk of fetal toxicity and malformations [23].

### 3.2. Preeclampsia

PE is defined as new-onset hypertension with proteinuria or hypertension and relevant organ dysfunction (with or without proteinuria) after 20 weeks of gestation [106].

The exact pathway by which PE increases the risk of CVD remains unclear. Several hypotheses have been proposed: increased likelihood of abnormal lipid deposition within the spiral artery, altered vascular remodeling such as an atherosclerotic process, oxidative stress, and inflammatory response caused by narrowing of the lumen of spiral arteries, which may persist beyond pregnancy and contribute to this to vascular dysfunction [107].

A meta-analysis by McDonald et al. has been evaluated the risk of CVD in 116,175 women with a PE or eclampsia compared with women with normal pregnancies. A previous diagnosis of PE was associated with a nearly twofold increased risk of CV and cerebrovascular complications and CV mortality [108].

### 3.3. Gestational Diabetes (GD)

GD has been closely associated with a higher risk of developing CV disease. Nurse’s Health Study (NHS) II, at a follow-up of 25 years, an increase of 43% of CVD complications risk (AMI or stroke) has been well recognized in women with a previous GD diagnosis [109]. GD seems to predispose to developing a subsequent type 2 DM. Indeed in NHS II, DM was observed in 19% of women with documented GD compared with 4.8% of controls [108]. Some possible pathophysiological mechanisms include a reduction in coronary flow reserve (CFR), early atherosclerosis, and endothelial dysfunction [110]. These findings suggest that primary prevention of CVD should be implemented early in the postpartum period [111].

### 3.4. Polycystic Ovary Syndrome(PCOS)

Polycystic ovary syndrome (PCOS) includes oligomenorrhea, an excess of androgens, infertility [112], and insulin resistance (IR) involving 6–10% of all women in the reproductive age [113,114]. Women with PCOS have an increased risk of developing hypertension and diabetes during pregnancy [114].

A meta-analysis by Zhao involving 104, 392 women assessed a possible association between PCOS and CVD risk. Patients with PCOS were 1.3 times more likely to develop CVD than those without PCOS with a significantly increased risk of CAD [115]. Finally, in PCOS, the increased CVD risk and the development of hypertension and diabetes could be associated with a framework of MS, obesity, and IR [116]. However, these data remain still uncertain [115].

### 3.5. Autoimmune or Inflammatory Diseases

Autoimmune diseases (ADs) are more common in women [117]. The inflammatory state leads to endothelial dysfunction and accelerated atherosclerotic condition, causing premature cardiovascular CV events [118]. Moreover, cortisone treatment may exacerbate hypertension, diabetes, and hypercholesterolemia. A recent study involving patients with ADs confirms an increased risk for CVD and all-cause mortality [119].

### 3.6. Oral Contraceptives

Since the first COCs were introduced, the issue of CV adverse effects (i.e., thromboembolic events, stroke, AMI) has been widely discussed. The use of COCs seems to be associated with 2- to 4-fold greater relative risks of arterial and venous thromboembolic events. The risk of venous thromboembolism (VTE) related to COCs therapy in females aged <30 years is estimated to be 3.7/10,000 cases annually compared to 1.2/10,000 in subjects who do not use this therapy, and the risk increases considerably with age [114,120]. Even obese women taking COCs therapy may occur to increased risk of VTE [120]. COCs should not be prescribed for women with risk factors such as active or history of VTE and in stable clinical conditions regardless of anticoagulant therapy during major surgical procedures with expected long-term immobilization. The COCs are contraindicated in women with thrombophilic conditions, including factor V Leiden mutation, prothrombin G20210A mutation, protein C, protein S or antithrombin deficiency, as these factors furtherly increase the risk of VTE [121].

Nonetheless, a thrombogenic mutation in young, heterozygous females with no history of thromboembolism VTE has been assessed as a relative contraindication for hormonal contraceptives [120].

SLE with positive antiphospholipid antibodies (aPL) has an increased risk of arterial and venous thrombosis compared with normal subjects. If aPL are persistently positive, the risk of thromboembolism is further increased by the administration of COCs [121].

The overall risk of AMI and ischemic stroke is increased in women on COCs therapy in relation to dose [122,123,124]. The risk of MI and ischemic stroke does not vary with the type or generation of progestin [122]. The duration of therapy does not correlate to the risk of CV events and it disappears when the treatment is discontinued.

In women with migraines, the use of COCs causes about a 2-fold increase in the incidence of ischemic or hemorrhagic stroke [123]. COCs users with a history of migraine were 2 to 4 times as likely to have an ischemic stroke as non-users with a history of migraine. Migraines with aura are associated with higher stroke risk than those without aura [125,126].

Cigarette smoking provokes a 10-fold increase in AMI and a 2–3-fold increase in stroke incidence in COCs users. The use of COCs is related to increased nicotine metabolism and physiological stress response [127]. It is worthy to rule out hypertension during past pregnancies before COCs therapy is introduced in women with normal BP values. Women with a history of abnormal BP values during past pregnancies and use COCs later compared to COCs use with an adverse history of pregnancy-induced hypertension showed an increased risk of AMI and VTE. COCs may be adopted in women with well-controlled hypertension, aged ≤35, non-smokers with no additional conditions or symptoms of CVD [128].

In women >40 years of age with risk for AMI or stroke, progesterone-only COCs and levonorgestrel-releasing (IUDs) may be prescribed [129].

### 3.7. Menopause and Hormone Replacement Therapy (HRT)

Early menopause is identified with less than 40 years [130]. A strong correlation between menopause and CVD events has been well established and the incidence of CAD dramatically increases after menopause. Moreover, early-onset menopause leads to premature CAD [131]. Estrogens have a protective effect on the development of CAD [58,132]. On the contrary, women with an early onset of CAD (<35 years) are more likely to experience early menopause. After the first encouraging results showing that the hormone replacement therapy (HRT) after menopause reduced the incidence of CVD, subsequent studies have not confirmed the positive data for primary and secondary prevention. Remarkably, the Women’s Health Initiative study found that estrogens use in primary prevention was significantly associated with the risk of CV events compared with placebo [133]. Regarding secondary prevention, in a randomized trial conducted in women with established coronary artery disease CAD, treatment with conjugated equine estrogens plus medroxyprogesterone acetate has not reduced overall cardiac events and was documented an increase in thromboembolic events [134]. However, HRT with low-dose estrogens, or transdermal hormone therapy at the lowest possible dose with a short duration, is recommended in women less than ten years since the onset of menopause and <60 years. HRT should be individualized based on risk factors, the likelihood of having an ischemic event, and severe menopausal-related symptoms [135]. The HRT risk/benefit ratio debate continues with risks depending on the type, dose, duration of use, administration route, and initiation timing. Therefore, the treatment should be individualized using the best available evidence with periodic reevaluation [136]. Studies investigating HRT in women with post-menopausal hypertension have reported conflicting results. Transdermal estrogens replacement therapy is associated with a slight reduction of mean BP, suggesting beneficial effects. In women with AMI, menopausal HRT should be discontinued. A recent case-cohort study compared coronary event rates between 5486 premenopausal and 9916 postmenopausal women from 10 European countries [137]. No significant difference was found between postmenopausal and premenopausal women, although the age of menopause was related to a 2% increase in CV risk per year. In a recent analysis published, early menopause was identified as increasing the risk of non-fatal CVD before the age of 60 years. At the same time, this was not observed for women over 70 years of age [131]. Natural and surgical premature menopause was associated with an increased incidence of a composite endpoint of CAD, HF, aortic stenosis (AS), mitral insufficiency, atrial fibrillation, ischemic stroke, peripheral arterial disease, and VTE [138]. The Women’s Health Initiative randomized controlled trial included 16,608 postmenopausal women in primary prevention to HRT versus placebo; the study was closed prematurely after a follow-up of 5.2 years. Women taking the combined estrogen-progestin versus placebo had an increased risk of CVD, stroke, and elevated risk of breast cancer, concluding that the risks outweighed benefits [139]. In secondary prevention, conjugated equine estrogen plus medroxyprogesterone acetate did not reduce CV events in women with ischemic heart disease [134]. Thus, there is no indication for HRT in either primary or secondary prevention.

## 4. Gender Differences in Cardiovascular Diseases: From Epidemiology to Prognosis

### 4.1. Coronary Heart Disease (CAD)

CAD develops eight-ten years later in women than men, increasing after 55 in both genders [140]. Physiopathology, clinical presentation, the effectiveness of diagnostic tools, response to treatments, and outcomes profoundly differ between women and men [141]. A 2.6-fold higher incidence of CV events in postmenopausal compared with premenopausal has been shown in Framingham Study [141]. In Poland, the recent Polaspire study showed that women with CAD were older (*p* < 0.001) and with more multiple CV risk factors rate than men (*p* = 0.036) [142].

CAD causes death in the female gender more than any other pathology, and, surprisingly, the mortality rate for CAD is higher in women than in men [143,144]. However, acute coronary syndromes (ACS) incidence is higher in men than in women below 60 years, enhancing after 75 years in women [58].

Furthermore, for women aged 60 years or below presenting with ACS the mortality rate is almost 2-fold greater than same-aged men [145,146]. Several factors could explain this data. First of all, women with ACS symptoms are frequently more atypical than in men [58,147]. Secondly, women are more likely to delay clinical presentation than men [43,58].

Another reason is that women are less likely to receive acute reperfusion therapy [148,149] and optimal medical therapy (OMT) [148]. Finally, the female gender represents an independent risk factor for peri-procedural AMI and major bleedings after percutaneous coronary intervention, which is notably associated with increased mortality [58,150].

The anatomic coronary structure is also different between women and men. Notably, vessel sizes are smaller in women than men, and they have a greater systolic function but less diastolic compliance than men [43]. Moreover, in the female gender, coronary plaque erosion with distal embolization is more frequent, while plaque rupture with subsequent thrombosis is the typical angiographic finding in men [151,152]. Effectively, CAD in women is characterized by a more diffuse atherosclerotic burden often associated with microvascular and endothelial dysfunction, while epicardial coronary stenosis is typical of men [153]. Consistently, residual angina is more often present in the female gender after myocardial revascularization [154].

For all these anatomical features, ACS in the absence of CAD occurs more frequently in women than in men [154].

### 4.2. Takotsubo Syndrome (TS)

Takotsubo syndrome (TS) occurs in 1–2% of all patients referred for ACS with a prevalence of 7.5% of women [155], particularly in the postmenopausal period [156]. It is often provoked by psychological or physical stress [156].

TS has the clinical and electrocardiographic features of ACS without angiographically obstructive coronary artery disease. An acute and reversible (LVSD) and the reversible left ventricle apical ballooning are distinctive elements [156].

### 4.3. Spontaneous Coronary Artery Dissection (SCAD)

Regarding ACS, a peculiar etiology in women is spontaneous coronary artery dissection (SCAD), occurring in about 80% of cases in young and healthy women, 5% of which is related to pregnancy [157]. It is responsible for up to 25% of all ACS cases in women <50 years of age, presenting in about 75% with non-ST segment elevation myocardial infarction (NSTEMI) and 25% with ST-Segment Elevation Myocardial Infarction STEMI [157].

This phenomenon can be addressed to an artery wall structure frailty, and this vulnerability can be exacerbated by precipitating stressors (emotional or physical), which can trigger dissection [158]. The cause is thought to be either an intimal tear or bleeding from the vasa vasorum, resulting in an intramural hematoma. Pressure-driven expansion of the hematoma causes propagation of the dissected segment with the formation of a true lumen and a thrombus containing a false lumen [159]. Triggers for SCAD increase shear stress on the coronary artery wall, often mediated by elevated catecholamine levels and intra-abdominal pressure, like a Valsalva maneuver [160]. In this specific subset of CAD, conservative management is preferred in stable patients with SCAD as most dissected segments will heal spontaneously [161]. There is no evidence-based indication for treatment since no randomized clinical trials are available. Dual antiplatelet therapy with aspirin and clopidogrel is widely accepted, avoiding glycoprotein IIb/IIIa inhibitors (GPI) or fibrinolysis since they could delay arterial hematoma healing and potentially extend the dissection. BBs are recommended in all patients, with the potential to reduce arterial shear stress, facilitate healing and reduce long-term recurrence. It is reasonable to use statins as part of OMT in ACS for myocardial protection against ischemia, but the benefit in SCAD is unknown [162].

### 4.4. Ischemia with Non-Obstructive Coronary Arteries (INOCA)

INOCA has been recently recognized as a cardiac ischaemic condition without obstructive CAD, in which CMD added to epicardial coronary artery spasm causes the mismatch between blood supply and myocardial oxygen demands [163]. INOCA is more common in women, particularly those aged 45–65 years [163]. The prevalence of INOCA ranged from 62% of 883 female patients undergoing coronary angiography in Women’s Ischemia Syndrome Evaluation to 34.4% of 1022 women referred to CT angiography in ISCHEMIA trial [164]. Because of the elevated risk for major adverse CV events, an early diagnosis and treatment should be made [165]. Furthermore, ischemia is often non-related to obstructive CAD in women, so imaging tools detecting coronary stenosis could be unsuitable, causing failure or delayed diagnosis. Consequently, a specific treatment is not offered, increasing CV risk [58]. Novel diagnostic algorithms and a simplified classification of CMD could be helpful to properly accurately assess INOCA in women [34,163,166]. Nonetheless, further research is needed to recognize an adequate management strategy in women.

### 4.5. Myocardial Infarction in the Absence of Obstructive Coronary Artery Disease (MINOCA)

The diagnosis of MINOCA should be made following the Fourth Universal Definition of myocardial infarction (MI), requiring an AMI in the absence of obstructive CAD (no lesion ≥50%) [152]. MINOCA occurs in 5–6% of AMI [150]. A differential diagnosis from other cardiac and noncardiac disorders is essential [167]. Therefore, several diagnostic algorithms have been proposed, and cardiac magnetic resonance imaging (MRI) plays a vital role in excluding myocarditis and other cardiomyopathies [167]. Furthermore, provocative spasm testing, screening for thrombophilia conditions, and intravascular ultrasound could take part in the diagnostic process of MINOCA [167].

MINOCA patients are less likely to have dyslipidemia [152] and are often younger with only a slight male preponderance, although outcomes are similar for both sexes [162].

Although data on outcomes are contrasting [168,169] a considerable risk of non cardiac mortality seems to characterize MINOCA patients [170].

Therefore more accurate strategies are urgently required to assess the diagnosis and treatment of MINOCA, especially in women [58].

### 4.6. Heart Failure (HF)

HF is a growing pandemic health issue [171]. Although gender differences in epidemiology, etiology, clinical presentation, and outcomes have been emphasized for several decades, they are still not considered enough in clinical practice [152]. Furthermore, women still make up far less than 50% of study patients enrolled in HF clinical trials [172].

The overall estimated risk for HF is similar between men and women (around 20% at age 40 years and 30% at age 55 years) [172]. However, sex differences are significant for the type of HF affecting men and women (Figure 4). The prevalence of HFpEF is higher in women than in men [173], with an increase in the rate of women with HFpEF concerning women with reduced ejection fraction HH (HFrEF) [174].

In postmenopausal women, a shorter total reproductive duration seems to increase the risk of incident HF, suggesting a protective role of endogenous female sex hormones over one’s lifetime [175].

Pregnancy is a sex-specific risk factor for HF [176]. Peripartum cardiomyopathy is a condition that usually presents in the last weeks of pregnancy or the months following delivery [177]. Prompt diagnosis and appropriate management established by a multidisciplinary team are crucial for a favorable outcome of this condition [178]. Data analysis of clinical trials shows a higher rate of non-ischemic etiology in women than men (60% versus 43%, respectively) [179].

HF etiologies more common in women than in men include stress cardiomyopathy [156], which mainly affects postmenopausal women, and HF due to cardiotoxicity of some anti-cancer treatments, including drugs and radiation [180]. Furthermore, women with HF are more likely to have hypertension or valvular heart disease (VHA) as HF etiology, and generally are older and have a lower New York Heart Association (NYHA) functional class [181].

Overall, HF prognosis is better in women than in men [182]. Among HF patients with, LVSD, the female gender is associated with a longer survival time, and the survival difference between the genders is more prominent among non-ischemic patients [172]. Data on HF patients from the National Center for Health Statistics (NCHS), collected in the US from 2000 through 2014, show a higher death rate for men than for women in all age groups [183].

However, the female gender is associated with a worse health-related quality of life (HRQoL) in HFpEF and HFrEF [184,185]. Women hospitalized for HF have a longer hospitalization duration and are less likely to undergo procedures such as coronary angiography. However, they have a rate of in-hospital mortality similar to men [186]. Although clinical studies still do not include female patients consistently to HF prevalence, available data suggest equal benefit associated with the use of the four drugs able to impact HFrEF outcome and recommended by 2021 European Society of Cardiology (ESC) guidelines in both sexes [187]. However, observational studies [188,189] suggest that women may obtain the maximum clinical benefit with lower drug dosages recommended in international guidelines [187]. In patients with LVEF < 35%, women are less likely to undergo implantable cardioverter defibrillators (ICD) implantation than men [186] and seem to yield fewer benefits from ICDs [190]. However, women have a more favorable response to CRT than men in greater reverse cardiac remodeling and reduction of mortality and HF events [191,192].

Overall, due to the under-representation of women in randomized clinical drug trials, currently available sex-specific data are not sufficient to establish differences in the efficacy of the evidence-based pharmacological therapy in HF [193]. Focusing on sex differences in heart failure could eventually allow more individualized management.

### 4.7. Valvular Heart Diseases (VHD)

Relevant gender differences regarding the type of valve disease, pathology, clinical presentation, echocardiographic cut-off values, response to pharmacological therapy, surgical approach, and postoperative outcomes have been assessed [194]. However, there are no sex-related differences in the management of VHD.

In women, a higher prevalence of mitral valve (MV) prolapse [195] (often diagnosed in young women) [196], rheumatic [197] or degenerative MV stenosis [198], rheumatic mitral regurgitation (MR) [198] tricuspid valve (TV) stenosis and secondary TV regurgitation [198] has been well recognized [194]. On the contrary bicuspid aortic valve (AV) disease is less common in women. Still, moderate/severe aortic stenosis (AS), a smaller aortic annulus, a smaller LV outflow tract, and a more hypertrophied LV are often present in this condition [199].

Finally, valve fibrosis is predominant in degenerative AS with valvular calcification [198].

Gender-related anatomical and physiological diversities contribute to different hemodynamic responses in VHD [200].

Women are usually referred to surgery at a more advanced stage of valve disease than men because they require a longer time to reach the respective cut-off values for surgery treatment [201]. Moreover, in Transcatheter AV replacement (TAVR), periprocedural complications are higher in women than in men [202].

### 4.8. Atrial Fibrillation

Atrial fibrillation (AF) is the most common sustained arrhythmia in both sexes worldwide [203], representing one of the most relevant healthcare burdens in Europe [187]. The age-adjusted incidence of AF is generally 1.5 to 2 times lower in women than men. Still, the prevalence of women with AF is higher than men, especially in octogenarians, due to the greater longevity of [203].

Also, it has been demonstrated that women with AF are more symptomatic, have the worst quality of life, and recur to medical attention more frequently than men [204]. A significant difference according to sex in anticoagulation rates and time in therapeutic range (TTR) have not been assessed [204,205]. Furthermore, the AF prognosis seems to be slightly influenced by sex, although the results of different studies appear conflicting [203,204]. Finally, a lower risk of major bleeding events in females has been reported, although any difference in CV death rates, stroke, and systemic embolism has been found [206].

Catheter ablation (CA), a well-established treatment in AF, is less applied in women [206,207], and sex-based outcome studies have shown contradictory results [208].

In a recent large study using Nationwide Readmissions Database (NRD), authors compared sex-based outcomes, 30-day readmissions, and costs following catheter ablation of AF in more than 54,000 patients. They found that fewer female patients were referred to catheter ablation, and they were older with more comorbidities. Also, procedural complications were higher than men, even after adjusting for age, hospital factors, and comorbidities. Moreover body size, vascular anatomy, and the different response to anticoagulants could be responsible for the higher bleeding rate in women.

Moreover, the 30-day readmission rate for all-cause, cardiac cause, or recurrent AF was significantly more frequent in 25%, 48%, and 40% in female patients. Nevertheless, total cumulative costs for AF ablation were lower in female patients, probably due to reduced resource utilization [209].

Also, a sub-analysis of the Fire and Ice trial demonstrated that recurrent AF post-ablation was independently associated with the female sex, increasing the risk of recurrence up to 37% [210]. The higher rate of AF recurrence in women has been explained as a result of ablation as a late strategy in the course of the disease when the percentage of paroxysmal AF is lower and the rate of non-pulmonary vein triggers is increased. This may result from a different substrate in women, characterized by increased fibrosis [211].

These findings confirmed that women who suffered from the same arrhythmic disease as men have sex-specific risk factors, substrate, and outcome; thus, for this reason, they are treated differently, and the use of cardiac testing and procedures is significantly lower.

## 5. Treatment of Cardiovascular Risk Factors in Both Sexes: Differences and Relative Prognostic Impact

A gender disparity in CV risk factors has been found in clinical practice.

In the PHARMO database, the proportion of women using lipid-lowering drugs in primary prevention was lower than men [212]. A lower prescription of statin and ezetimibe in women has been also shown [213].

The achievement of the target values of BP and the regular intake of antihypertensive drugs, such as diuretics is more difficult to obtain in female patients. In contrast, men receive more often ACEI [25].

Recent studies, however, suggest an improvement in CV risk management in women [214].

Another question is whether cardiovascular risk factor management has the same impact on both sexes. The benefits of antihypertensive therapies have been revealed in both sexes [25], whereas the gender difference of these medications’ effects has not been well recognized [23]. Most studies evaluating the efficacy of statins have not been powered to compare the effectiveness between sexes specifically [64]. Statins have been found to have similar efficacy in lowering MACE and mortality in primary prevention irrespectively of sex [215]. The beneficial effect of aspirin use in women in primary remains controversial [47]. Low-dose aspirin did not prevent AMI in women in primary prevention, whereas it prevented ischemic stroke [216]. However, according to recent findings, should not be routinely used primary prevention [217,218,219]. Furthermore, an enhanced risk of bleeding and mortality has been associated with the use of aspirin in absence of MACE lowering [219,220,221].

A meta-analysis of 14 RCTs highlighted the important role of lifestyle education, counseling, and follow-ups, particularly in BP management, physical activity and blood glucose control [222].

Patients with HF in the most disadvantaged socioeconomic levels presented the worst degree of control for CV risk factors, and this negative effect was stronger in women [223].

However, guidelines for CV prevention are usually targeted to men rather than women; they allow to properly select type and dose of treatment in the male population [224]. Specific Gender-oriented guidelines have been proposed by American Heart Association in 2007 and modified in 2011 [225]; the risk assessment algorithm for women includes three categories, at high risk, at-risk, and with ideal CV health, based on the presence or absence of atherosclerotic vascular disease, major risk factors, subclinical atherosclerosis and healthy lifestyle [47,226]. Gender-specific guidelines can be a proper way to better treat CV risk factors in women [227].

## 6. Treatment of Cardiovascular Diseases in Both Sexes: Differences and Relative Prognostic Impact

In spite of the high incidence of CVD, more significant difficulties to healthcare access and gender disparities in treatment are commonly experienced by women (Figure 5). There are many observational reports of under-treatment of women for stable CAD and suspected or diagnosed ACS [228]. They receive less cholesterol screening, less evidence-based treatment, and less counseling [229].

### 6.1. Under-Treatment

In the PHARMO database, drug use in secondary prevention was lowest in young women [212]. Women with stable CAD had poorer control of CV risk, and ideal body weight, BP, LDL-C, and hemoglobin A1c (HbA1c) levels are often non-well-controlled. 

In the Australian Health Survey, among participants (n=11,518 patients, aged 45–74), only 21.8% of women received the OT compared to 41.4% of men [230]. A lower prescription of based-evidence CV drugs in women with CAD resulted from the National Health and Nutrition Examination Survey (NHANES) [231]. Underuse of statin therapy in women with CVD was reported by the analysis of the Department of Veterans Affairs (VA) administrative data [232]. Furthermore, female sex has been correlated with lower use of statin therapy in age-adjusted analysis from the U.S. Medical Expenditure Panel Survey (MEPS) on patients with atherosclerotic CVD [233]. These findings have been confirmed by a retrospective analysis on 16,898 (26% females) and 71,358 (49% females) patients, aged <65 years and >66 years respectively within one-month follow-up after MI [234].

What is more, in spite of the confirmed effectiveness of statins in both sex, not only statin therapy is less likely to be prescribed in women, but also a minor use of high-intensity versus low-intensity statins has been described in females compared to men in secondary prevention [234]. Lower adherence to treatment in females has been proposed as a possible reason [235].

The risk of ACS in females has generally underestimated [236]. On one hand, precise diagnosis in women is often delayed and, on the other hand, they are less likely to receive appropriate medical treatment (BBs, ACEI or ARBs, statins, aspirin, and P2Y12 antagonists) and invasive interventions [1,215]. In The Netherlands, in both STEMI and NSTEMI, use of aspirin, P2Y12-inhibitors, statins, BBs, ACEI/ARBs, vitamin-K antagonists, or novel oral anticoagulants (NOACs) was higher in male patients [237]. Women less frequently received other quality indicators compared with men, including timely percutaneous coronary interventions (PCI) or thrombolytic therapy for STEMI and timely coronary angiography for NSTEMI, and coronary artery bypass grafting (CABG), and this is not explained by differences in baseline characteristics [215,238,239,240,241,242] A possible reason explaining the undertreatment of women with CAD could be that they have significantly more angiographically normal coronary arteries or non-significant CAD in STEMI and NSTEMI [243].

A greater delay in referring women to Emergency Medical Service (EMS) has been also recorded in the Victorian Cardiac Outcomes Registry [244]. Besides, residual symptoms of angina negatively impact the HRQL’ of female patients with CAD [215].

Angina is frequently misdiagnosed in women, particularly in absence of coronary arteries lesions [245]. Female sex was associated, instead, with the prescription of short- and long-acting nitrates [47].

Furthermore, women have been less frequently referred for carotid endarterectomy, (CA), and heart transplantation (HT) than men with the same recognized indications [58]. Finally, in Peripheral Arterial Disease (PAD) women are less likely to receive appropriate therapy, especially in older age [246]. Women also receive fewer ICD than men [247].

### 6.2. Cardiac Rehabilitation

Women are less likely to be involved in disease-modifying programs for cardiac secondary prevention than men. Data analysis on 297,719 patients (45.0% of men and 38.5% of females referred for CR) showed that the enrollment of women in Cardiac Rehabilitation (CR) is significatively lower in women than men (*p* < 0.00001) [248]. CR adherence has been also reported poorer in women [249].Women’s lesser referrals and participation in cardiac rehabilitation (CR) programs are widely recognized, with multilevel barriers including higher age, comorbidities, sedentary, lower education level, lack of social support, and increased burden of family responsibilities [250].

However, it seems that women attending CR have better outcomes [251].

Nonetheless, because men and women are non-identical, a critical question is whether these gender-based variations in treatment correspond to lower-quality care.

### 6.3. Gender Disparities in Adherence and Responses to Treatments

Female gender is significantly associated with medication non-adherence among CV patients [237,252].

Moreover, women’s responses to CV treatments are heavily influenced by differences in physiologic features, which lead to strong disparities in drug pharmacokinetics (absorption, distribution, metabolism, and physiologic features and pharmacodynamics [253]. Women experience more relevant adverse effects and pharmacotoxicity with the most frequently used CV drugs, such as Torsade de Pointes (TdP) because of QT-prolonging drugs, cough, and, ACE inhibitor-induced angioedema, statin intolerance, and more bleeding complications [252,254]. Some medications require a particular dose adaptation in women: BBs (metoprolol), CCBs (verapamil), loop, and thiazide diuretics. Trends towards a more significant benefit for women have been observed, in contrast, for endothelin receptor antagonists (ERAs), CCBs (amlodipine), ACEI (ramipril), and aldosterone antagonists (eplerenone). Pregnancy and labor present challenges in increased thromboembolic and bleeding risk and COCs promote platelet aggregation. Several data show greater basal and residual on-treatment platelet reactivity in females [255]. In real life, however, women with NSTEMI are more likely to receive excessive antithrombotic treatments. Furthermore, a higher risk of in-hospital bleedings and access-related complications after PCI occurs in women [256].

The effects of clopidogrel, prasugrel, and ticagrelor seem to be comparable in both sex [215,255].

The Gp IIb/IIIa inhibitors use has been associated with more adverse effects and less favorable outcomes in women [215].

The benefit of antiplatelet therapy should be carefully assessed evaluating drawbacks such as bleedings. Moreover preventing from over-dosage of antithrombotic treatment related to different values of BMI, and renal function should be mandatory in clinical practice [257].

According to current guidelines, antithrombotic therapies should be considered for prevention in patients irrespective of gender [255]. However, weight and renal function should be rigorously assessed in anti-platelet prescription to lower the risk of bleeding in female [258].

Finally, as has it has been recommended by ESC, equal timely and intensively access to care and treatments should be guaranteed to women who are referred for NSTEMI compared to men [256].

NOACs seem to be more effective in women, with a lower risk of bleeding, in women, whereas no gender differences have been for warfarin use in the correct TTR categories [256,259].

In secondary prevention trials, in general, all the antiplatelet and anticoagulant medications are equally effective reducing mortality and MACE, although women tend to experience a higher relative benefit due to their poorer risk profile [215].

With regards to the use of ACEI and ARBs, they share similar effects in both sexes, with small differences in favor of more noticeable positive effects in males according to post-hoc analysis of clinical trials [215,254]. BBs showed similar positive results in both sexes, with a slightly more survival benefit in old-women [215,254]. The original Digitalis Investigation Group trial demonstrated a lack of mortality benefit in HFrEF, and subsequent retrospective analyses suggested increased mortality among women [215,254]. While the proportion of women included in trials with spironolactone and eplerenone was small, non-significant interaction terms between treatment effect and sex suggest similar efficacy among women [215,254]. The beneficial effect of valsartan/sacubitril were shown in both sexes for the composite outcome, but adverse events were not analysed stratified by sex and in the PARADIGM trial only one-fifth of the sample as women [254].

Clinical trial data to support the treatment of HFpEF, the predominant form of HF among women, are scarce and generally neutral [215]. Just as concerning is that HF trials were not powered to detect sex–drug interactions, to test the benefit in the subgroup of women, or to identify a drug that would only be effective in women [254].

### 6.4. Outcomes

Outcomes after therapeutic interventions may differ between genders [215,252,253,254,260,261] (Table 1).

Higher 30-day mortality and major bleedings have been reported in women with STEMI compared with men [244]. Among STEMI patients, a significantly higher risk of bleeding after fibrinolysis has been observed in women [215]. Historically, previous studies have reported a relation between female gender and adverse outcomes (all-cause mortality and MACE) after PCI. In a systematic review and meta-analysis on 49 studies, involving 1,032,828 patients, the in-hospital and at least 2-years mortality with CAD after PCI was significantly lower in males [260]. The same benefits from PCI in women as men have been reported. In most contemporary studies, there were no significant gender differences in the mean number of vessels attempted, lesions treated, stents used, or in use of drug eluting stents. So far, after PCI female gender is an independent predictor for peri-procedural AMI and major bleedings; moreover, greater short-term morbidity and mortality have been assessed [215,252]. Due to the smaller size of female coronary arteries, a radial approach should be preferred in women to reduce peri-procedural bleedings [215].

Transradial-PCI, however, was associated with a lower risk of major bleedings, mortality, and MACE irrespective of gender [262]. The 30-day readmission rates for HF and recurrence of events after ACS than men are higher in women than in men.

Besides, women are more likely to develop adverse events and consequently, discontinuation of therapy is more frequent [215]. Moreover, low-dose ionizing radiation exposure from interventional procedures risk of cancer seems to be more significant in women.

A similar outcome after elective PCI has been reported in both sexes although worse results in females have been previously described [215,259].

In contrast to the majority of CV procedures [58], a greater reduction of morbidity and mortality in women has been observed in transcatheter aortic valve replacement (TAVI) trials [251].

When considering the effectiveness of ICD in women with HF, pooled data showed no benefit in the prevention of sudden cardiac death (SCD). A meta-analysis also demonstrated that women have more device-related complications as death, all-cause readmissions, and HF readmissions. At 1-year there MACE and mortality risk after ICD implantation was similar in both sexes [261]. Moreover, women with HFrEF and a left-bundle branch block undergoing CRT, seem to have more benefit in terms of more pronounced mortality reduction, accompanied by a remarkable reverse remodeling of LV (Figure 6) [196].

## 7. Conclusions

CVD is the leading cause of mortality in women, causing about one-third of female deaths [44]. Women have been used to incorrectly considered “protected” against CVD with a poor awareness in patients and physicians so that scarce prevention of CV events is frequent, and treatment strategies should be extensively improved [44]. Governments and health institutions could operate several interventions to reverse this trend.

Careful screening and aggressive management of risk factors should be adopted to reduce CV events, favoring lifestyle measures and adherence to treatments in women. A better knowledge of sex-related differences is crucial to tailored CV therapeutic approaches in men and women.

## Figures and Tables

**Figure 1 jcm-11-01176-f001:**
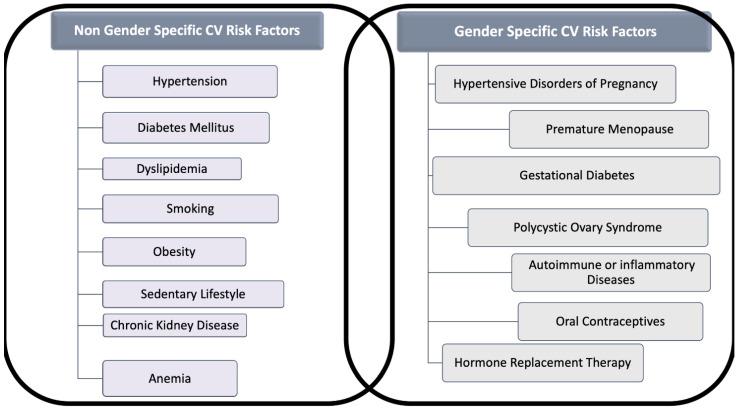
Gender and Non-Gender Cardiovascular (CV) Risk factors for cardiovascular disease in women. The figure distinguishes risk factors for cardiovascular disease in two categories: (A) Those that are related to gender, often under-recognized and (B) those that interest both sexes, but which might act in women differently than in men.

**Figure 2 jcm-11-01176-f002:**
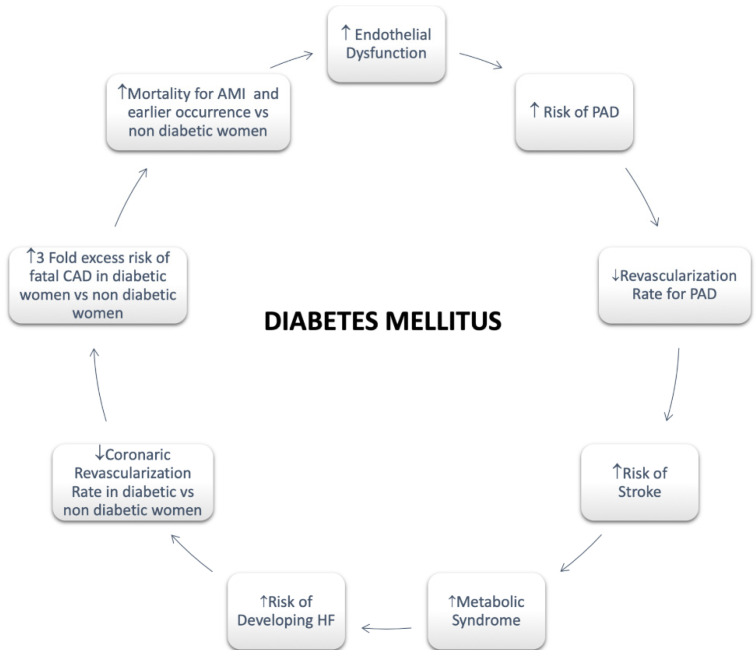
Diabetes Mellitus in Women. Abbreviations: Peripheral artery disease; HF: Heart Failure; CAD: Coronary Artery Disease; AMI: Acute Myocardial Infarction.

**Figure 3 jcm-11-01176-f003:**
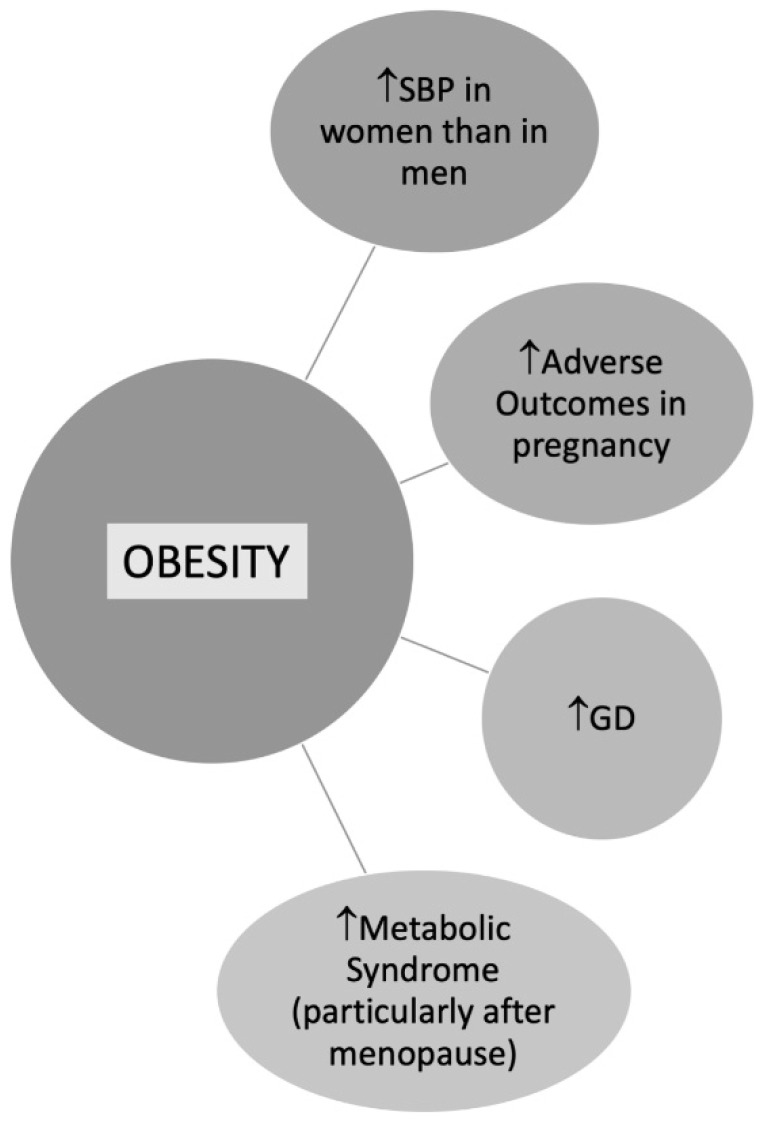
Obesity in Women. Abbreviations: SBP: Systolic blood pressure; GD: Gestational diabetes.

**Figure 4 jcm-11-01176-f004:**
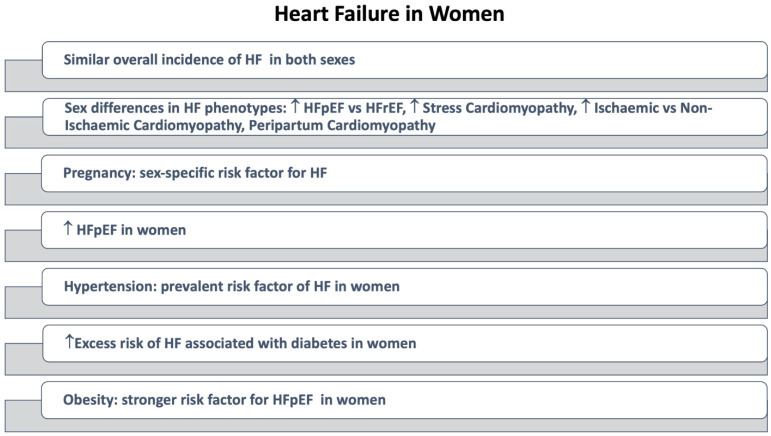
Heart Failure in Women. Abbreviations: HF: Heart Failure, HFpEF: Heart Failure with preserved ejection fraction; HFrEF: Heart failure with reduced ejection fraction.

**Figure 5 jcm-11-01176-f005:**
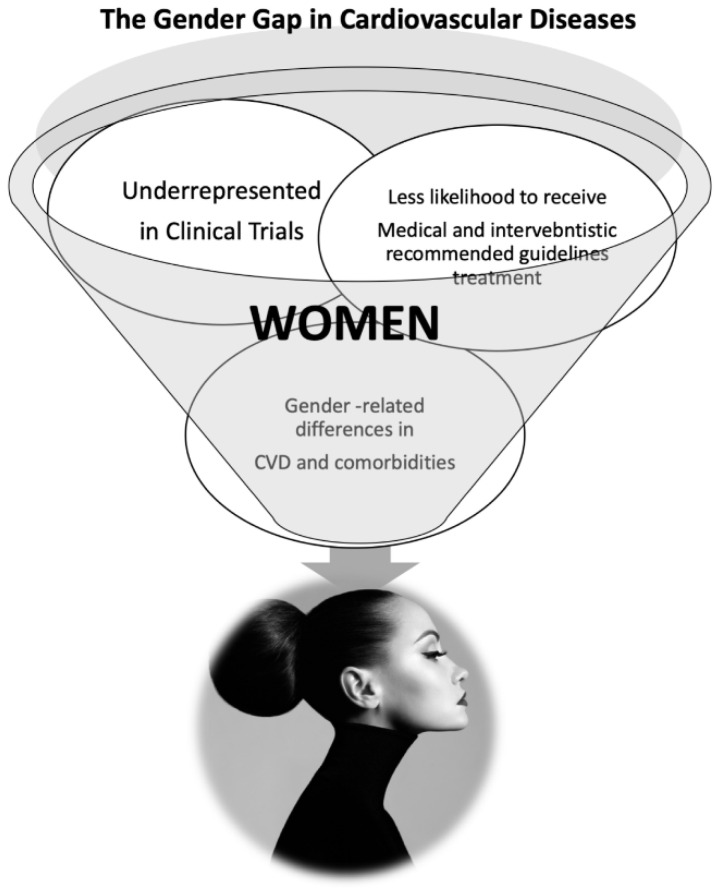
The gender GAP in CVD.

**Figure 6 jcm-11-01176-f006:**
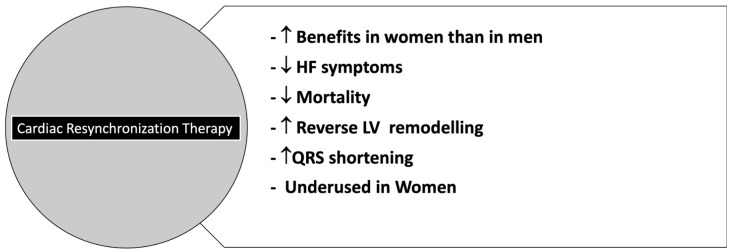
CRT in Women. Abbreviations: HF: Heart Failure; LV: left Ventricle.

**Table 1 jcm-11-01176-t001:** Effects of treatment in women on clinical outcomes in different cardiovascular settings compared with men.

Contest	Effects
ACS in Women	STEMI: ↑30-day mortality ↑ Bleeding [244]
	After PCI: ↑Peri-procedural AMI ↑Major Bleeding ↑In-Hospital Mortality ↑2-years Mortality [208,244]
	Gp IIb/IIIa inhibitors: ↑Adverse effects ↓ Outcomes [208]
Stable CAD in Women	After elective PCI: Similar contemporary outcomes in both sex [208]
Atrial Fibrillation and Stroke	NOACs: ↑Effectiveness ↓ Bleeding Risk
	Warfarin in the correct TTR: No differences in CV mortality all-cause mortality and stroke risk between the sexes
	In stroke: Thrombolysis, antiplatelet and anticoagulant therapies ↑ beneficial
Heart Failure	With comparable treatment: ↑Clinical Outcomes [208,246].
	BBs: ↑ Slightly survival benefit in elderly [208,246].
	Digitalis: ↑Mortality [208,246]
	Spironolactone Eplerenone: Similar efficacy [208,246].
	Valsartan/Sacubitril: Equal beneficial effect [246]
	After CV procedures: ↑Morbidity ↑Mortality ↓ Health-related quality of life ↓ Functional improvement [58]
Surgical, Interventional, and Electrophysiology Treatments	TAVI: Beneficial in terms of short-, mid-, and long-term outcomes [243]
	ICD in women: ↑Device-related complications ↑Death ↑All-Cause readmissions ↑HF readmissions
	CRT: ↓ Mortality ↑Reverse remodeling of LV [183]
Cardiac Rehabilitation	↑ Outcomes [255]

Abbreviations: ACS: Acute Coronary Syndrome, STEMI: ST-Elevation Myocardial Infarction, PCI: Percutaneous Coronary Intervention, AMI: Acute Myocardial Infarction, Gp: Glycoprotein, NOACs: Novel Oral Anticoagulants, TTR: Target Therapeutic Range, CV: Cardiovascular, BBs: Beta-blockers, TAVI: Transcatheter Aortic Valve Implantation, ICD: implantable cardioverter-defibrillator, CRT: Cardiac resynchronization therapy, LV: Left Ventricle.

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
