# Peer review of "Update on Management of Cardiovascular Diseases in Women"

_jcm, 2022, doi:10.3390/jcm11051176_

Round 1
Reviewer 1 Report
This paper reviewed the main cardiovascular risk factors in women related to CVD and discussed the current gender differences in CVD treatment. The motivation behind the problem investigated in this manuscript is interesting and meaningful. However, there are some problems to be further improved as well.
- The references cited were a little old, and the proportion of references within 5 and 3 years should be increased before the article can be accepted.
- In 2.1.2, the target organ injury were discribed in heart, kidney and brain without eyes, which need to be added.
- In 2.5,“women with obesity are also at higher risk of ...mental disorders." is this part necessary for this paper?
- The abbreviation should be marked when the word first appears, such as, in page 6, “CAD”.
- There is at least one writing error in the manuscript, such as, in page 8, “ between 5486 and 9916 postmenopausal women ” would be “between 5486 premenopausal and 9916 postmenopausal”. Please check the manuscript carefully.
- In part 6, "tretment of cardiovascular diseases in both sexes: differences and relative prognositic impact": it is recommended to set some sections to respectively describe the underdiagnosis in female, the undertreatment in female (drugs, intervention, poor compliance), different responses to treatment, the understudy, etc.
Author Response
Reviewer 1
This paper reviewed the main cardiovascular risk factors in women related to CVD and discussed the current gender differences in CVD treatment. The motivation behind the problem investigated in this manuscript is interesting and meaningful. However, there are some problems to be further improved as well.
1) The references cited were a little old, and the proportion of references within 5 and 3 years should be increased before the article can be accepted.
-We agree with the reviewer. We changed the oldest references and we added more recent ones. A few articles published in 2022 have been also cited.
2) In 2.1.2, the target organ injury was described in the heart, kidney, and brain without eyes, which need to be added.
-We agree with the reviewer. We added Hypertensive Retinopathy.
3) In 2.5, “women with obesity are also at higher risk of ...mental disorders." is this part necessary for this paper?
- We agree with the reviewer. We deleted this sentence because it was not necessary.
4)The abbreviation should be marked when the word first appears, such as, in page 6, “CAD”.
- We checked the abbreviations and we corrected them in the test (Coronary Heart Disease, Heart Failure etc)
5) There is at least one writing error in the manuscript, such as, in page 8, “ between 5486 and 9916 postmenopausal women ” would be “between 5486 premenopausal and 9916 postmenopausal”.Please check the manuscript carefully.
-We corrected this error.
6) In part 6, "treatment of cardiovascular diseases in both sexes: differences and relative prognostic impact": it is recommended to set some sections to respectively describe the underdiagnosis in female, the undertreatment in female (drugs, intervention, poor compliance), different responses to treatment, the understudy, etc.).
-We agree with the reviewer. We divided this part into sections.
Reviewer 2
This review compromises relevant risk factors and outcome parameters of cardiovascular disease in male vs. female patients. The review is interesting and raises an updated topic of great importance. However, several changes are needed before publication.
7) The cited guidelines for blood pressure treatment are aged. Please refer to current CV prevention guidelines.
- We changed this reference with the latest one.
8) Regarding organ damage, please refer to microvascular obstruction in women as well.
-We agree with the reviewer and we better explained microvascular obstruction in the text
9) Regarding figure 1: please re-organize so that the raised aspects are in the same order as it is read throughout the text.
- We reorganized the figure one
10) Please add a graphical abstract or more general figure containing main aspects that are raised in the review.
-We added 5 other general figures.
11) Table 1 needs better structure. It is hard to follow through. The typing is very small and you get lost. Reorganize and focus on key points if you would like to present it in a table.
- We modified and reorganized the table.
12) Regarding passage 6. – does the number of participating patients in disease-modifying programs for secondary prevention differ in male vs. female patients?
-We agree with the reviewer, we better explained that women are less likely to be involved in disease-modifying programs for cardiac secondary prevention than men and we cited a meta-analysis on 297,719 patients
13) Page 9/27 – what do you mean with the numbers in round brackets?
-We agree with the reviewer. We deleted the number in the brackets because they were unclear.
14) Please refer to further comorbidities that are increased in female patients as well – e.g. CKD or anemia. This might influence outcome as well. Please add.
-We agree with the reviewer. We added a section on CKD and a section on anemia.
15) As rate of T2DM is higher in female patients, cellular alterations might influence cardiac outcome especially in women. Please refer to PMID: 34171330 We agree with the reviewer.
- We added reference “Dannenberg L, Weske S, Kelm M, Levkau B, Polzin A. Cellular mechanisms and recommended drug-based therapeutic options in diabetic cardiomyopathy.Pharmacol Ther. 2021;228:107920.10.1016/j.pharmthera.2021.107920. Epub 2021 Jun 24. PMID: 34171330
16) Please complement information for Euroaspire IV in the text (number of patients, year)
-We agree with the reviewer. We added complement information on Euroaspire IV in the text. We also mentioned the EUROASPIRE V survey.
17) Page 2/27 – blood pressure units are missing.
-We added mmHg
18) In 2.3 (cholesterol) – Impact of age and menopause in women is too short. Please explain more precisely.
-We agree with the reviewer and we better explained the impact of age and menopause in women in the texts.
18) The authors explain at several points that CAD risk is low in pre-menopausal women and increases with age. Please give concrete numbers and give information if rise is even higher in women compared to men.
-We agree with the reviewer. We gave more precise information in the text.
19) Passage 6 is very long. Please separate in sections.
-We agree with the authors and we separated passage 6 into sections.
20) Headlines of passages have different typing. Please adapt.
-We adapted headlines of passages.
21) Table of content in the beginning to have a first overview would be helpful. -We added a table of content.
22) Both paragraphs about treatment could be in the end for better structure. -We agree with the reviewer.
-We put paragraphs about treatment at the end of the paper.

Reviewer 2 Report
This review compromises relevant risk factors and outcome parameters of cardiovascular disease in male vs. female patients. The review is interesting and raises an uptodate topic of great importance. However, several changes are needed before publication.
- The cited guidelines for blood pressure treatment are aged. Please refer to current CV prevention guidelines.
- Regarding organ damage, please refer to microvascular obstruction in women as well
- Regarding figure 1: please re-organize so that the raised aspects are in the same order as it is read throughout the text.
- Please add a graphical abstract or more general figure containing main aspects that are raised in the review.
- Table 1 needs better structure. It is hard to follow through. The typing is very small and you get lost. Reorganize and focus on key points if you would like to present it in a table.
- Regarding passage 6. – does the number of participating patients in disease modifying programs for secondary prevention differ in male vs. female patients?
- Page 9/27 – what do you mean with the numbers in round brackets?
- Please refer to further comorbidities that are increased in female patients as well – e.g. CKD or anemia. This might influence outcome as well. Please add.
- As rate of T2DM is higher in female patients, cellular alterations might influence cardiac outcome especially in women. Please refer to PMID: 34171330
- Please complement information for Euroaspire IV in the text (number of patients, year)
- Page 2/27 – blood pressure units are missing
- In 2.3 (cholesterol) – Impact of age and menopause in women is too short. Please explain more precisely.
- The authors explain at several points that CAD risk is low in pre-menopausal women and increases with age. Please give concrete numbers and give information if rise is even higher in women compared to men.
- Passage 6 is very long. Please separate in sections.
- Headlines of passages have different typing. Please adapt.
- Table of content in the beginning to have a first overview would be helpful.
- Both paragraphs about treatment could be in the end for better structure.
Author Response
Reviewer 2
This review compromises relevant risk factors and outcome parameters of cardiovascular disease in male vs. female patients. The review is interesting and raises an updated topic of great importance. However, several changes are needed before publication.
1) The cited guidelines for blood pressure treatment are aged. Please refer to current CV prevention guidelines.
- We changed this reference with the latest one.
2) Regarding organ damage, please refer to microvascular obstruction in women as well.
-We agree with the reviewer and we better explained microvascular obstruction in the text
3) Regarding figure 1: please re-organize so that the raised aspects are in the same order as it is read throughout the text.
-We reorganized figure one
4) Please add a graphical abstract or more general figure containing main aspects that are raised in the review.
-We added 5 other general figures.
5) Table 1 needs better structure. It is hard to follow through. The typing is very small and you get lost. Reorganize and focus on key points if you would like to present it in a table.
.- We modified and reorganized the table.
6) Regarding passage 6. – does the number of participating patients in disease-modifying programs for secondary prevention differ in male vs. female patients?
-We agree with the reviewer, we better explained that women are less likely to be involved in disease-modifying programs for cardiac secondary prevention than men and we cited a meta-analysis on 297,719 patients
7) Page 9/27 – what do you mean with the numbers in round brackets?
-We agree with the reviewer. We deleted the number in the brackets because they were unclear.
8) Please refer to further comorbidities that are increased in female patients as well – e.g. CKD or anemia. This might influence outcome as well. Please add.
-We agree with the reviewer. We added a section on CKD and a section on anemia.
9) As rate of T2DM is higher in female patients, cellular alterations might influence cardiac outcome especially in women. Please refer to PMID: 34171330 We agree with the reviewer. - We added reference “Dannenberg L, Weske S, Kelm M, Levkau B, Polzin A. Cellular mechanisms and recommended drug-based therapeutic options in diabetic cardiomyopathy.Pharmacol Ther. 2021;228:107920.10.1016/j.pharmthera.2021.107920. Epub 2021 Jun 24. PMID: 34171330
10) Please complement information for Euroaspire IV in the text (number of patients, year)
-We agree with the reviewer. We added complement information on Euroaspire IV in the text. We also mentioned the EUROASPIRE V survey.
11) Page 2/27 – blood pressure units are missing.
-We added mmHg
12) In 2.3 (cholesterol) – Impact of age and menopause in women is too short. Please explain more precisely.
-We agree with the reviewer and we better explained the impact of age and menopause in women in the texts.
13) The authors explain at several points that CAD risk is low in pre-menopausal women and increases with age. Please give concrete numbers and give information if rise is even higher in women compared to men.
-We agree with the reviewer. We gave more precise information in the text.
14) Passage 6 is very long. Please separate in sections.
-We agree with the authors and we separated passage 6 into sections.
15) Headlines of passages have different typing. Please adapt.
-We adapted headlines of passages.
16) Table of content in the beginning to have a first overview would be helpful. -We added a table of content.
17) Both paragraphs about treatment could be in the end for better structure. We agree with the reviewer. -
We put paragraphs about treatment at the end of the paper.
Round 2
Reviewer 2 Report
The authors satisfactorily addressed all aspects.